# Comparison of pre-treatment with different diluted sufentanil in reducing propofol injection pain in gastrointestinal endoscopy: A randomized controlled study

Qian Su[1☺], Shu He[2☺], Lu Li[1], Xiangqing Wei[1], Xin Sun[1], Xuefeng Yang[1], Boxiang Du[1‡], Lei Yao [1‡*]

1 Department of Anesthesiology, Nantong First People's Hospital, Nantong, Jiangsu, China, 2 Department of Epidemiology and Biostatistics, School of Public Health, Nantong University, Nantong, Jiangsu, China

☺ These authors contributed equally to this work.
‡ LY and B-XD also contributed equally to this work and should be considered co-corresponding author, and LeiYao should be acknowledged as the main corresponding author.
* yaolei1987wisdom@sina.com

## Abstract

### Background

This study aimed to investigate the efficacy of pre-treatment with different concentrations of sufentanil in mitigating propofol injection-induced pain.

### Methods

This study included 421 patients who were scheduled for gastrointestinal endoscopy between June 2023 and December 2024. Participants were randomly assigned to one of the four groups with different concentrations sufentanil: 0 µg/mL group(0.9% normal saline), 0.5 µg/mL group, 1 µg/mL group and 5 µg/mL group.

### Results

Among the four groups, the rates of pain relief were 82 (77.4%), 93 (89.4%), 96 (89.7%), and 91 (87.5%), respectively. Compared to the 0 µg/mL group, the other groups demonstrated significantly reduced pain following propofol injection (p < 0.05). The recovery times were 14.59 ± 3.92 min, 15.13 ± 3.20 min, 14.27 ± 3.06 min, and 15.57 ± 3.24 min, respectively. Notably, the 1 µg/mL group did not exhibit a prolonged recovery time compared to the 0 µg/mL group. The total propofol consumption was recorded as 218.5 ± 36.8 mg, 196.7 ± 31.0 mg, 183.8 ± 25.0 mg, and 189.6 ± 31.4 mg, respectively, with the 1 µg/mL group showing the lowest total propofol consumption among the groups. The incidences of adverse events (AEs) were 61.3%, 70.2%, 58.9%, and 76.9%, respectively. In comparison to the 1 µg/mL group, the 5 µg/mL

**Data availability statement:** Data are available from the The Second Affiliated Hospital of Nantong University Ethics Committee (contact via Dr. Hongqing Xu) for researchers who meet the criteria for access to confidential data. His contact details were xuhongqing000@126.com and +86 159 5131 2678. After the acceptance of the manuscript for publication, we will release our raw data in the NGDC Data at the following link: https://ngdc.cncb.ac.cn/omix/preview/tzPvhtFn.

**Funding:** Q.S. was supported by the Health Commission of Nan-tong City Health and Family Planning Scientific Research Projects (QNZ2023026) and The High-Level Scientific Research Project Cultivation Fund of Nantong First People's Hospital (YPYJJYB24008). L.Y was supported by the Health Commission of Nan-tong City Health and Family Planning Scientific Research Projects (MS2024033) and Nantong Municipal Science and Technology Scientific Research Projects (MSZ2024097).

**Competing interests:** The authors have declared that no competing interests exist.

group exhibited a higher incidence of AEs. Furthermore, multivariate analysis indicated that a 5 µg/mL dilution of sufentanil increases the risk of AEs (p < 0.05).

## Conclusions

The 1 µg/mL group demonstrated greater safety and efficacy when combined with propofol.

## Trial registration

The study has been registered in the Chinese Clinical Trial Registry (ChiCTR). Link of the registry: http://www.chictr.org.cn. Date of registration: 2023/06/12. Trial registration number: ChiCTR2300072402.

---

## Introduction

Propofol is the most commonly used intravenous anesthetic for both the induction and maintenance of anesthesia. Propofol-induced pain is a prevalent complication of anesthesia, with an incidence rate of 28% to 90% [1]. Although this pain is transient, the intense discomfort it causes can trigger trigger preoperative anxiety, reduce future medical compliance, and impair patient satisfaction. Additionally, pain-induced sympathetic activation increases myocardial oxygen demand, particularly hazardous for cardiovascular patients [2]. Furthermore, involuntary movements during endoscopic procedures may necessitate additional propofol doses, elevating complication risks. Consequently, various medications, including opioids, ketamine, and lidocaine, have been employed to prevent propofol-induced pain [3–5].

In addition to medications, various factors may be involved in the development of propofol-induced pain, such as the injection site, vein size, injection speed and propofol temperature [6–9]. Sufentanil has been demonstrated to significantly mitigate the injection pain associated with propofol [6,10]. It is the most commonly used adjunct administered with propofol for painless gastrointestinal endoscopy in China [11]. Sufentanil's high potency, which is 5–10 times greater than that of fentanyl, creates a narrow therapeutic window. While it is effective against injection pain, dose escalation paradoxically increases the risks of respiratory depression and hypoxemia [12,13]. Therefore, identifying an optimal concentration of sufentanil that can effectively alleviate injection pain while minimizing adverse effects is of paramount importance.

In classic pharmacokinetics theory, after intravenous (IV) administration of sufentanil, the drug enters systemic circulation almost immediately. In the bloodstream, approximately 93% of sufentanil binds to plasma proteins; only the unbound fraction, nearly 7%, is capable of crossing biological membranes to reach target sites, such as the brain and spinal dorsal horn [14]. However, this model has significant limitations when applied to rapid IV push scenarios. For instance, the rapid influx of drugs into the systemic circulation can lead to altered distribution kinetics that are not accounted for in conventional pharmacokinetic assessments [15]. Additionally, physiological

changes such as diabetes and hypertension, due to disease states, can further complicate the prediction of drug metabolism. The chronic inflammatory state often observed in diabetic patients may affect hepatic and renal drug metabolism, leading to altered pharmacokinetic profiles for many medications [16]. Hypertension can also impact the distribution of drugs within the body. Increased blood pressure can lead to enhanced perfusion of tissues, which may alter the volume of distribution (Vd) for certain medications. For example, drugs that are highly protein-bound may experience changes in their free fraction due to alterations in plasma protein levels, such as albumin, which can be affected by both hypertension and its associated conditions [17]. This extended administration correlates with gradual plasma concentration increases and attenuated peak levels. Despite these mechanistic insights, no clinical study has systematically evaluated the differences in analgesic effects and adverse effects of different dilutions of sufentanil across different populations.

To fill these knowledge gaps, we prospectively investigated the associations between various concentrations of sufentanil in saline and propofol injection-induced pain. This study aimed to identify an optimal concentration of sufentanil that effectively alleviates injection pain while minimizing adverse effects, which is of paramount importance. The primary outcomes of the study was the pain relief rate, secondary outcomes were recovery time, total propofol consumption and adverse events.

## Participants and methods

### Participants, study design and treatments

This research was carried out at the Second Affiliated Hospital of Nantong University, located in Nantong City, Jiangsu Province, China. The study period extended from March 1, 2023, to December 31, 2023, and encompassed participants aged 18–65 who voluntarily underwent a non-invasive gastrointestinal endoscopy.

The exclusion criteria were as follows:

1.  Patients with severe bronchopulmonary disease, cardiovascular conditions (e.g., heart attack, coronary heart disease, stroke, and congestive heart failure), neurological disorders [18] (e.g., multifocal epilepsy, comorbid neurological diseases, or severe psychiatric disorders), and hepatic encephalopathy (HE) as defined by the West Haven criteria. In addition, patients with severe renal disease (CKD stages 4 and 5) [19] were excluded from this study;

2. Patients who had allergy to propofol or sufentanil;

3. Patients with a preoperative assessment of difficult airway or BMI ≥ 30 kg/m$^2$;

4. Patients who suffered from chronic pain or took analgesics for a long term;

5. Patients who underwent puncture for larger antecubital vein;

6. Patients who needed to undergo endoscopic treatment of polyp or a diagnostic ultrasound gastroscopy;

7. Patients who declined to participate;

8. Patients with bradycardia (HR < 60 bpm) [20].

### Randomization and blinding

EpiCalc 2000 software was used to produce the randomization schedule. Participants were randomly assigned to one of the four groups in a 1:1:1:1 ratio using a computer-generated randomization schedule. Each participant received a distinct identification number. Data were collected for analysis after anonymization.

In this study, the drug preparation process was blinded using the "envelope" method. Based on the randomization sequence inside the envelopes, a research nurse with no involvement in the clinical care of patients prepared three different concentrations of sufentanil (Yichang Humanwell Pharmaceutical Co., China) by diluting it with 0.9% saline: 5 μg/mL

for the 5 µg/mL group (1 ml total), 1 µg/mL for the 1 µg/mL group (5 ml total), and 0.5 µg/mL for the 0.5 µg/mL group (10 ml total). The 0 µg/mL group received 10 ml of 0.9% normal saline. 20 ml of 1% propofol was purchased from Liaoning Haisco Pharmaceutical Co., China.

Considering that researchers were blinded, different volumes of drugs were provided identically in opaque 10 ml injectors. Two drugs, propofol and sufentanil, were given separately by two anesthesiologists. Patients and researchers, including gastroenterologists and anesthetists, were blinded to the group location.

### Trial intervention

On arrival at the examination room, a 07-model intravenous infusion needle (specification parameter 0.7 mm×25 mm) was inserted into a vein on the dorsum of the patient's right hand. We monitored pulse oximetry, ECG (lead II), and non-invasive blood pressure. The intravenous medication was given by two individuals. All patients will receive supplemental oxygen at a flow of 2–4 L/min through a nasal cannula.

Thirty seconds after when one anesthesiologist (who was blinded to participants' allocation) administered the pretreatment drug, the other anesthesiologist administered intravenous 1% propofol (initial dose of 2.5 mg/kg) at a rate of 0.5 ml/sec. If patients involuntarily moved during the examination, another dose of propofol was given at a dose of 0.5 mg/kg [21]. The pharmaceutical agents were administered to patients with overweight (defined as BMI ≥ 25 kg/m²) based on lean body weight (LBM) [22,23]. The anesthesiologist assessed the degree of pain every 5 seconds using the Numerical Rating Scale (NRS) with a 3-point system: 0 for no pain, 1 for mild pain, 2 for moderate pain, and 3 for severe pain [24]. In the case of desaturation or unplanned low blood pressure, physicians elevated the mandible or used ephedrine (6 mg iv.) as rescue treatment. All adverse events were recorded during gastrointestinal endoscopy.

### Measurement

Pain relief rate: This metric indicates the proportion of patients achieving an NRS score of 0 or 1, indicating those who experience significant pain alleviation. The adverse events included hypoxia, cough, hypotension, bradycardia, dizziness, nausea, and vomiting (Specific definitions are provided in the S1 Table). These events were chosen because they necessitated specific interventions and/or had significant implications.

### Sample size calculation

This study used the Chi-Square test in PASS 15.0 software to determine the sample size. Based on our pilot results, we divided patients into four groups: the saline group, the 5 µg/mL group, the 1 µg/mL group, and the 0.5 µg/mL group. We categorized the pain severity into four levels: no pain, mild pain, moderate pain, and severe pain. Consequently, the degrees of freedom for the test were determined to be 9. The total effective rates (including no pain and mild pain) for the four groups were 70%, 85%, 90%, and 85%, respectively. Therefore, the effect size (W value) was calculated to be 0.1974. With a Type I error rate of 0.05 in a two-sided test, and a Type II error rate of 0.2, the required sample size was determined to be 402. Taking into account a dropout rate of 15%, 464 participants were included, with 116 participants in each group.

### Statistical analysis

Statistical analysis was conducted using SPSS 27.0, and PASS15.0. Measurement data conforming to the Gaussian distribution are presented as mean ± SD, and categorical data are expressed as numbers or percentages (%). Continuous data were analyzed using one-way analysis of variance. Categorical data were analyzed using the Chi-square test or Fisher's exact test. Z test was used for further comparison between the two groups. One way ANOVA was used for comparison between multiple groups, and the LSD-t test was used for further comparison between two groups. The pain scores (NRS) were compared among the four groups using the Kruskal-Wallis test, and a Pairwise test was used for

further comparisons between the two groups. All statistical tests were two-tailed, and the corrected differences were considered significant when p < 0.05.

## Trial oversight

The trial was conducted following the Declaration of Helsinki and the trial protocol was approved on February 25, 2023 by the Ethics Committee of The Second Affiliated Hospital of Nantong University, Reference: 2023KT006). This study was registered in the Chinese Clinical Trial Registry (ChiCTR2300072402, website: http://www.chictr.org.cn). All patients provided written informed consent before participation.

## Results

### Characteristics of the study population

In total, 464 patients were initially enrolled. Forty-three individuals were excluded due to different reasons, leaving 421 patients for inclusion (Fig 1).

Demographic data are presented in Table 1. Overall, there were no significant differences among the groups in terms of age, sex, weight, body mass index, ASA physical status, duration of examination, medical history of hypertension or diabetes, and smoking or drinking habits (Table 1; p > 0.05).

### Pain relief rate

After propofol injection, significant differences were observed between the four experimental groups in terms of pain score (see Fig 2 and Table 2). Among the four experimental groups, the rates of participants reporting mild pain were 34.9%,

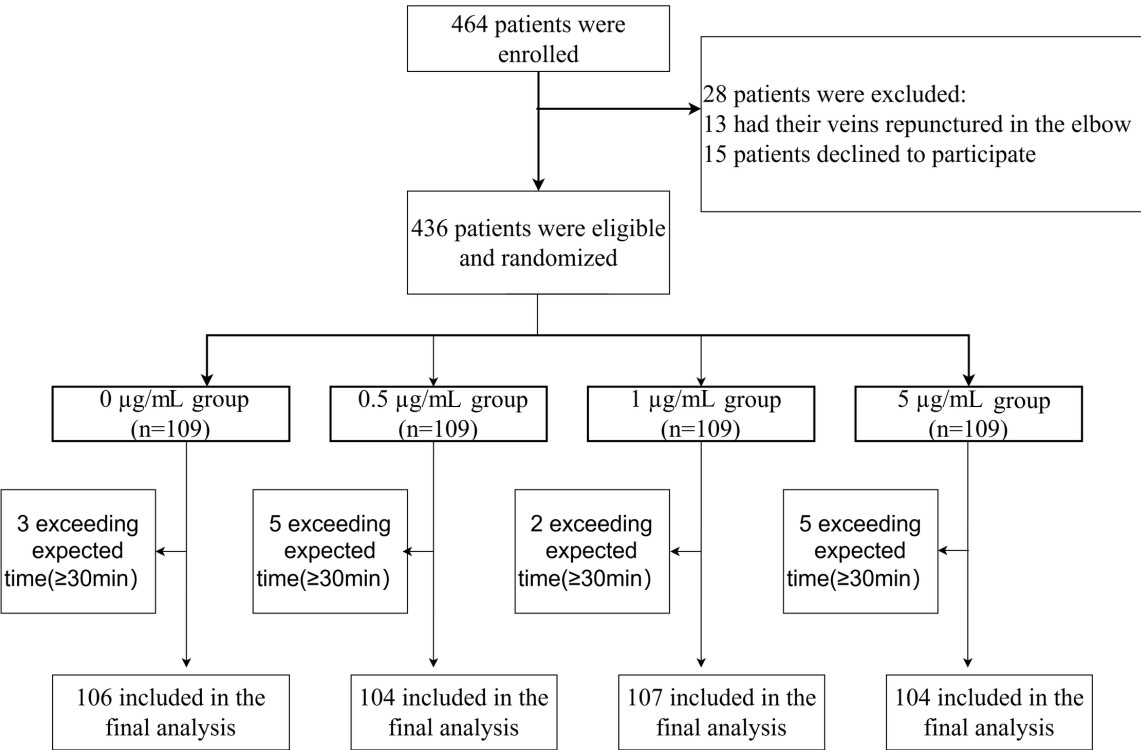

**Fig 1. Chart of flow.**

**Table 1. Baseline participant characteristics.**

| Characteristics | 0µg/mL group (n = 106) | 0.5µg/mL group (n = 104) | 1µg/mL group (n = 107) | 5µg/mL group (n = 104) | P value |
|---|---|---|---|---|---|
| Age(y) | 44.5±10.4 | 45.1±10.6 | 44.3±10.5 | 45.2±10.4 | 0.904[a] |
| Sex(male/female) | 58/48 | 51/53 | 55/52 | 53/51 | 0.872[b] |
| Weight (kg) | 66.2±10.7 | 66.3±10.5 | 67.6±10.7 | 66.8±12.1 | 0.782[a] |
| Body mass index (Kg/m²) | 23.5±2.7 | 23.8±2.6 | 24.1±2.7 | 24.0±2.7 | 0.402[a] |
| <18.5(n, %) | 3(2.8%) | 2(1.9%) | 3(2.8%) | 3(2.9%) | |
| 18.5~24.9(n, %) | 72(67.9%) | 69(66.3%) | 62(57.9%) | 63(60.6%) | |
| 25~29.9(n, %) | 31(29.2%) | 33(31.7%) | 42(39.3%) | 38(36.5%) | |
| ASA physical status (n, %) | | | | | |
| I | 51(48.1%) | 50(48.1%) | 48(44.9%) | 43(41.3%) | 0.222[b] |
| II | 29(27.4%) | 35(33.7%) | 41(38.3%) | 30(28.8%) | |
| III | 26(24.5%) | 19(18.3%) | 18(16.8%) | 31(29.8%) | |
| Smokers (n, %) | 25(23.6%) | 19(18.3%) | 17(15.9%) | 27(26.0%) | 0.247[b] |
| Drinkers (n, %) | 35(33.0%) | 27(26.0%) | 24(22.4%) | 28(26.9%) | 0.372[b] |
| Hypertension (n, %) | 23(21.7%) | 15(14.4%) | 21(16.9%) | 29(27.9%) | 0.119[b] |
| Diabetes (n, %) | 4(3.8%) | 5(4.8%) | 2(1.9%) | 6(5.8%) | 0.469[c] |
| Duration of the examination(min) | 12.0±4.4 | 12.1±3.7 | 12.1±3.6 | 12.1±4.6 | 0.998[a] |

[a]: F-test (one-way analysis of variance);

[b]: Pearson Chi-Square;

[c]: Fisher's test

Data adhering to a normal distribution are presented as mean±SD. Data for ordinal categorical variables are represented as the number (or percent) of cases.

25.0%, 27.1%, and 31.7%, respectively. The rates of participants reporting moderate pain were 18.8%, 7.7%, 9.4%, and 11.5%, respectively. The rates of participants reporting severe pain were 3.8%, 2.9%, 0.9%, and 1.0%, respectively. The rates of pain relief (NRS scores of 0 or 1 point) were 77.4%, 89.4%, 89.7%, and 87.5%, respectively.

Pairwise comparisons revealed statistically significant differences between the 0 µg/mL group and both the 0.5 µg/mL group (p = 0.004) and the 5 µg/mL group (p = 0.007), whereas no significant intergroup variations were observed in other comparisons (P > 0.05, Table 3).

### Recovery time

The recovery time is defined as the duration from the conclusion of the endoscopy to the point at which patients regain consciousness and open their eyes independently, as observed and recorded by the researcher (see Fig 2 and S2 Table). The recovery times observed in the study were as follows: the 0 µg/mL group exhibited a recovery time of 14.59±3.92 minutes, the 0.5 µg/mL group recorded a recovery time of 15.13±3.20 minutes, the 1 µg/mL group showed a recovery time of 14.27±3.06 minutes, and the 5 µg/mL group had a recovery time of 15.57±3.24 minutes(S2 Table). Compared to the 0 µg/mL group, there was no significant difference in recovery time between the 0.5 µg/mL group and the 1 µg/mL group (p > 0.05). However, the 5 µg/mL group exhibited a longer recovery time when compared to both the 0 µg/mL group and the 1 µg/mL group (p < 0.05, see S3 Table).

### Total propofol consumption

Total propofol consumption is defined as after the examination, the researcher records the total amount of propofol used. The total propofol consumption in the 0 µg/mL group was 218.5±36.8mg. The total propofol consumption in the 0.5 µg/mL

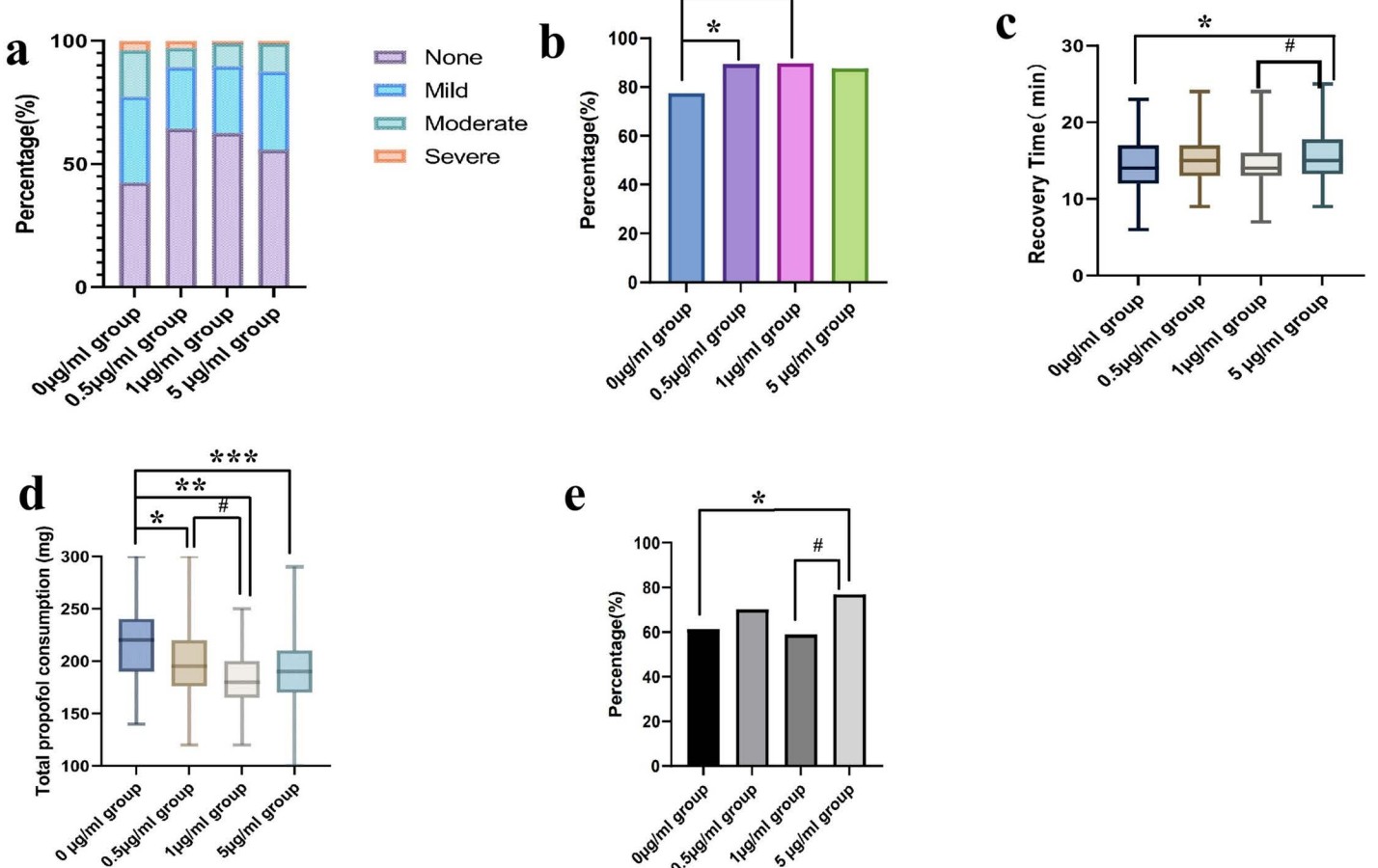

**Fig 2. The main outcomes among four groups. (a) The severity of pain. (b) The rate of pain relief.**\*p < 0.05 vs. 0 µg/mL group, \*\*p < 0.05 vs. 0 µg/mL group **(c) The recovery time.**\*p < 0.05 vs. 0 µg/mL group,# p < 0.05 vs. 1 µg/mL group).**(d)The total propofol consumption.**\*p < 0.05 vs. 0 µg/mL group, \*\*p < 0.05 vs. 0 µg/mL group, \*\*\*p < 0.05 vs. 0 µg/mL group, #p < 0.05 vs. 1 µg/mL group.**(e) The incidence of adverse events.**\*p < 0.05 vs. 0 µg/mL group, #p < 0.05 vs. 1 µg/mL group.

**Table 2. Frequency and percentage distribution of pain intensity grades across four experimental groups.**

| Degree of Pain, n(%)* | 0µg/mL group (n = 106) | 0.5µg/mL group (n = 104) | 1µg/mL group (n = 107) | 5µg/mL group (n = 104) |
|---|---|---|---|---|
| None (0 point) | 45 (42.5) | 67 (64.4) | 67 (62.6) | 58 (55.8) |
| Mild pain (1 point) | 37 (34.9) | 26 (25.0) | 29 (27.1) | 33 (31.7) |
| Moderate pain (2 point) | 20 (18.8) | 8 (7.7) | 10 (9.4) | 12 (11.5) |
| Severe pain (3 point) | 4 (3.8) | 3 (2.9) | 1 (0.9) | 1 (1.0) |
| Pain relief rate (NRS ≤ 1) | 82(77.4) | 93 (89.4) | 96 (89.7) | 91 (87.5) |

SF: Sufentanil; Pairwise P: Pairwise comparison P values;

*The pain scores (NRS) were compared among the four groups using the Kruskal-Wallis test, H = 14.655, P = 0.002.

group was 196.7±31.0 mg. The total propofol consumption in the 1 µg/mL group was 183.8±25.0 mg. The total propofol consumption in the 5 µg/mL group was 189.6±31.4 mg(as shown in S4 Table). Compared to the 0 µg/mL group, the total propofol consumption significantly decreased in the other three groups (p < 0.05, S5 Table). Furthermore, compared to the 0.5 µg/mL group, we observed a significantly lower total propofol dose in the 1 µg/mL group (p < 0.05).

The regression analysis demonstrated a linear association between total propofol consumption and recovery time (slope = 0.063, $R^2$ = 0.398, p < 0.001, Fig 3a). Specifically, the analysis indicated that for every 1 mg/kg increase in total propofol consumption, the average recovery time was extended by 0.063 minutes.

**Table 3. Bonferroni-adjusted pairwise comparisons of pain intensity grades among four experimental groups.**

| Pairwise comparisons | Standard Error | Raw P Value | Adjusted P Value |
|---|---|---|---|
| 0 µg/mL vs. 0.5 µg/mL | 14.962 | <0.001 | **0.004** |
| 0 µg/mL vs. 1 µg/mL | 14.856 | 0.001 | **0.007** |
| 0 µg/mL vs. 5 µg/mL | 14.962 | 0.024 | 0.143 |
| 0.5 µg/mL vs. 1 µg/mL | 14.927 | 0.870 | 1.000 |
| 0.5 µg/mL vs. 5 µg/mL | 15.033 | 0.264 | 1.000 |
| 1 µg/mL vs. 5 µg/mL | 14.927 | 0.337 | 1.000 |

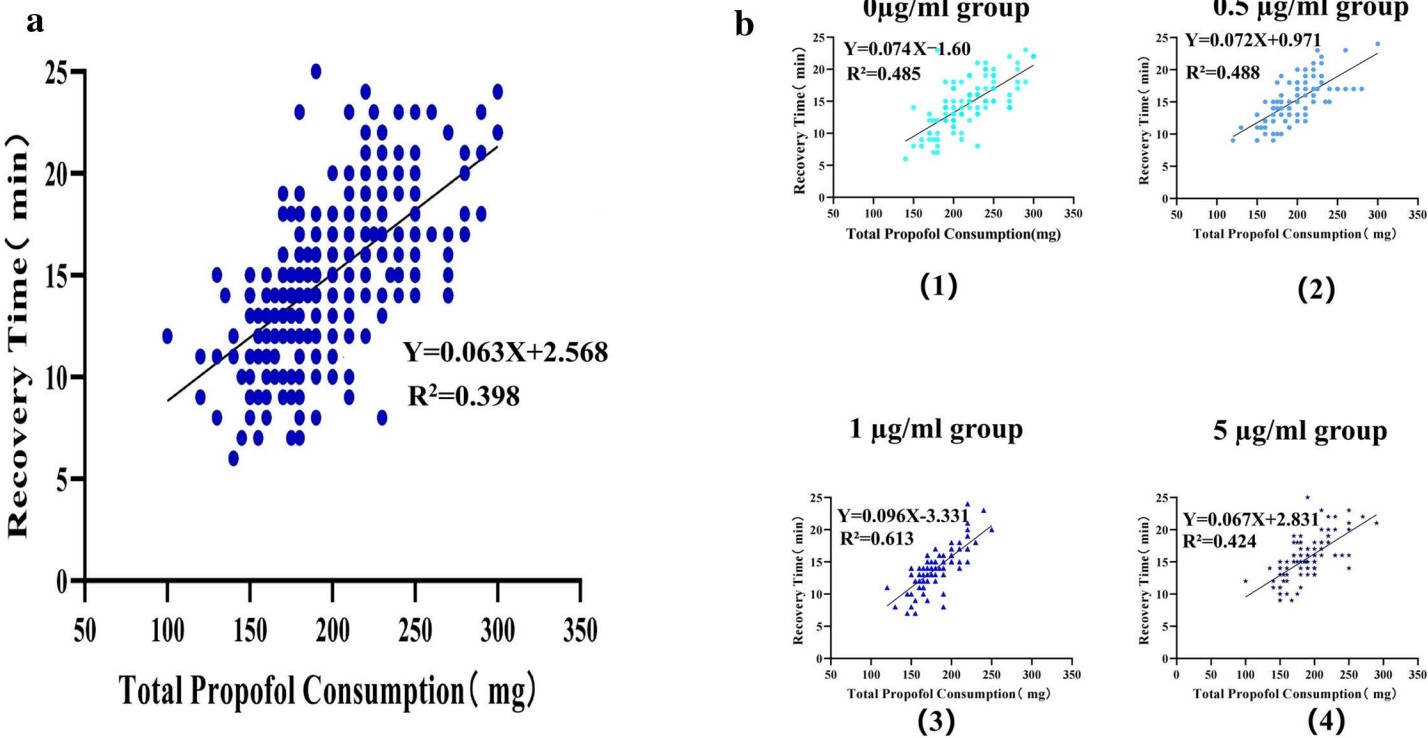

**Fig 3. Association between total propofol consumption and recovery time. (a)** Scatter plot demonstrating the correlation between total propofol consumption (x-axis) and recovery time (y-axis). $R^2$ represents coefficient of determination. **(b)** Stratified analysis by sufentanil groups, with separate regression lines fitted for each group. Colors/symbols distinguish subgroups, and regression equation are provided for each model.

Stratified linear regression by sufentanil concentration groups (Fig 3b) revealed varying association strengths. The 0 µg/mL group exhibited a modest positive association (b=0.074, p<0.001), while the 0.5 µg/mL and 1 µg/mL groups showed similar (b=0.072, p<0.001) and slightly stronger (b=0.096, p<0.001) associations, respectively. In contrast, the 5 µg/mL group had a weaker association (b=0.067, p<0.001), suggesting that higher sufentanil concentrations may be linked to a reduced relationship between total propofol consumption and recovery time.

## Adverse events

After the gastrointestinal endoscopy was completed, significant differences were observed between the four groups in terms of the incidence rates of adverse events (Table 4). The 0 µg/mL group had an incidence of 61.3% for adverse events. The 0.5 µg/mL group had an incidence of 70.2% for adverse events. The 1 µg/mL group had an incidence of 58.9% for adverse events. The 5 µg/mL group had an incidence of 76.9% for adverse events.

The most frequent negative effect was hypotension (27.6%), occurring in 16.0% of individuals in the the 0 µg/mL group, 40.4% in the 0.5 µg/mL group, 24.3% in the 1 µg/mL group, and 29.8% in the 5 µg/mL group (p<0.05). Compared to the 0 µg/mL group, the 0.5 µg/mL, 1 µg/mL, and 5 µg/mL groups exhibited a significantly higher incidence of hypotension (p<0.05, Fig 4).

The second most common adverse event was dizziness (24.2%), occurring in 28.3% of individuals in the the 0 µg/mL group, 24.0% in the 0.5 µg/mL group, 13.1% in the 1 µg/mL group, and 31.7% in the 5 µg/mL group(p<0.05). Compared to the 0 µg/mL group, the 1 µg/mL group showed a significantly lower incidence of dizziness (28.3% vs. 13.1%, p<0.05). Compared to the 0.5 µg/mL group, a significant reduction in the incidence of dizziness was observed in the 1 µg/mL group (24.0% vs.13.1%, p<0.05). Further analyses indicated that dizziness were significantly less frequent in the 1 µg/mL group compared to the 5 µg/mL group (31.7% vs. 13.1%, p<0.05).

The incidence of cough among individuals in the 0 µg/mL group was 16.0%, in the 0.5 µg/mL group was 4.8%, in the 1 µg/mL group was 0.9%, and in the 5 µg/mL group was 6.7 (p<0.05). Compared to the 0 µg/mL group, the 0.5 µg/mL group showed a significantly lower incidence of cough (4.8% vs. 16.0%, p<0.05). Compared to the 0 µg/mL group, the 1 µg/mL group exhibited a significantly lower incidence of cough (0.9% vs. 16.0%, p<0.05). Similarly, the 5 µg/mL group demonstrated a reduced rate of cough compared to the 0 µg/mL group (6.7% vs. 16.0%; p<0.05). Furthermore, a significant reduction in the incidence of cough was observed in the 1 µg/mL group compared to the 0.5 µg/mL group (0.9% vs. 4.8%, p<0.05).

The incidence of hypoxia among groups was 10.4%, 15.4%, 15.9% and 23.1%, respectively. The incidence of brady-cardia among groups was 10.4%, 24.0%, 19.6% and 15.4%, respectively. The incidence of nausea, and vomiting among

**Table 4. The incidences of adverse events among four groups.**

| Incidence n(%) | 0µg/mL group (n=106) | 0.5µg/mL group (n=104) | 1µg/mL group (n=107) | 5µg/mL group (n=104) | P Value |
|---|---|---|---|---|---|
| Hypoxia | 11 (10.4) | 16 (15.4) | 17 (15.9) | 24 (23.1) | 0.096[a] |
| Cough | 17 (16.0) | 5 (4.8) | 1 (0.9) | 7 (6.7) | **<0.001[b]** |
| Hypotension | 17 (16.0) | 42 (40.4) | 26 (24.3) | 31 (29.8) | **<0.001[a]** |
| Bradycardia | 11 (10.4) | 25 (24.0) | 21 (19.6) | 16 (15.4) | 0.057[a] |
| Dizziness | 30(28.3) | 25(24.0) | 14(13.1) | 33(31.7) | **0.010[a]** |
| Nausea or Vomiting | 2(1.9) | 1(1.0) | 1(0.9) | 0(0) | 0.808[b] |
| The Incidence Rates of AEs* | 65(61.3) | 73 (70.2) | 63 (58.9) | 80 (76.9) | **0.020[a]** |

*The Incidence Rates of AEs:refers to the frequency at which patients experience at least one adverse event during a study.

[a]: Chi-square test;

[b]: Fisher's exact test.

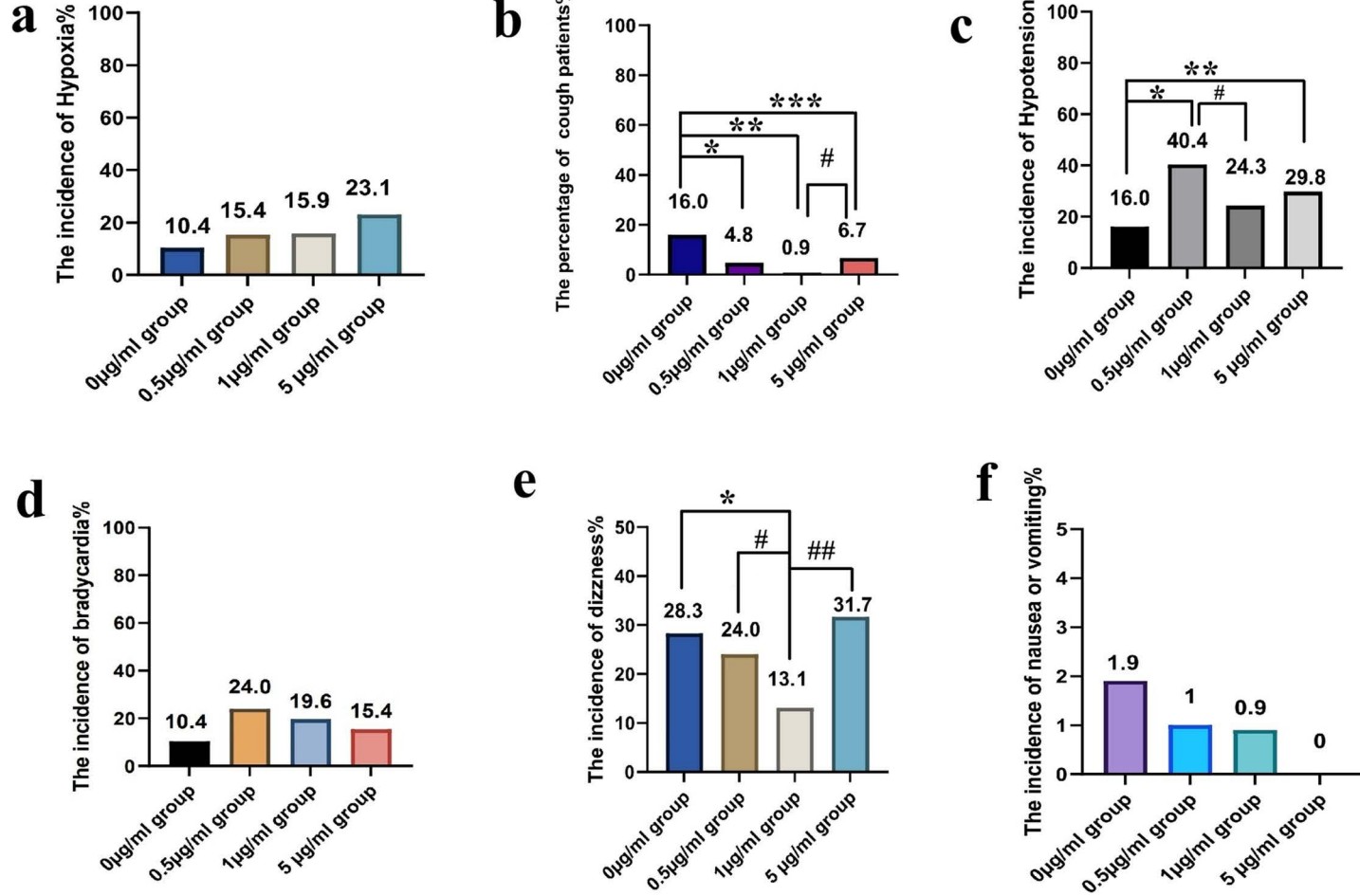

**Fig 4. The Incidences of adverse events among the four groups.** Post hoc pairwise comparisons were performed with Bonferroni-adjusted chi-square tests (α = 0.0083). **(a) The incidence of hypoxia. (b) The incidence of cough.** *p < 0.05 vs. 0 μg/mL group, **p < 0.05 vs. 0 μg/mL group, ***p < 0.05 vs. 0 μg/mL group, #p < 0.05 vs. 1 μg/mL group.**(c) The incidence of hypotension.** *p < 0.05 vs. 0 μg/mL group, **p < 0.05 vs. 0 μg/mL group, #p < 0.05 vs. 1 μg/mL group. **(d)The incidence of bradycardia. (e)The incidence of dizziness.** *p < 0.05 vs. 0 μg/mL group, #p < 0.05 vs. 0.5 μg/mL group, ##p < 0.05 vs. 5 μg/mL group. **(f) The incidence of nausea or vomiting.**

groups was 1.9%, 1.0%, 0.9% and 0%, respectively. However, there were no significant differences in the incidence of hypoxia, bradycardia, nausea, and vomiting among four groups (p > 0.05).

## Multivariable analysis

The results of the multiple linear regression analysis indicated that the regression equation was significant, with an F-value of 46.408 and a p-value of less than 0.001 (see Table 5). Among the analyzed variables, total propofol consumption (β = 0.778, p < 0.001) and sufentanil concentration at varying dilution levels ((β = 0.330, p < 0.001) were significantly positively correlated with recovery time. Conversely, body mass index (BMI) (β = −0.088, p < 0.05) exhibited a negative correlation with recovery time. Notably, age, sex, hypertension, diabetes, smoking, and alcohol history were not significant predictors of the time to awakening. Collectively, these variables accounted for 49.3% of the variance in recovery time.

**Table 5. Multiple linear regression between the recovery time and demographic and health risk factors.**

| Variable | Unstandardized B | Standardized Coefficients β | t | P Value | F | Adjusted R Square |
|---|---|---|---|---|---|---|
| Total Propofol Consumption | 0.078 | 0.778 | 19.662 | **< 0.001** | 46.408 | 0.493 |
| Age | 0.018 | 0.055 | 1.451 | 0.148 | | |
| Gender | −0.588 | −0.087 | −1.927 | 0.055 | | |
| BMI | −0.112 | −0.088 | −2.288 | **0.023** | | |
| Hypertension | −0.009 | −0.001 | −0.028 | 0.978 | | |
| Diabetes | 0.680 | 0.040 | 1.109 | 0.268 | | |
| Drinkers | 0.238 | 0.031 | 0.779 | 0.436 | | |
| Smokers | −0.065 | −0.008 | −0.193 | 0.847 | | |
| Sufentanil Groups* | 1.002 | 0.330 | 8.834 | **< 0.001** | | |

*Sufentanil groups refer to different concentrations of diluted sufentanil.

Multivariable logistic regression analysis identified a sufentanil concentration of 5 µg/mL as the sole significant predictor of adverse events (adjusted OR = 2.07, 95% CI [1.13, 3.80], p = 0.018, Table 6). When compared to controls, this concentration threshold was associated with a 2.07-fold increase in the risk of adverse event occurrence (p < 0.05).

## Interaction analysis

The results indicated that the interaction term "total propofol consumption × sufentanil group" was not statistically significant (p > 0.05, Table 7). This indicates that the relationship between propofol dosage and adverse effects does not vary significantly across the sufentanil subgroups. Clinical implications suggest that propofol's contribution to adverse effects remains consistent regardless of the intensity of sufentanil concentration, indicating that the risks are additive rather than synergistic.

## Discussion

In this study, we investigated the safety and pain relief efficacy of various dilutions of sufentanil pretreatment in gastrointestinal endoscopy. A concentration of sufentanil at 1 µg/mL demonstrated a significant advantage over other concentrations, resulting in a higher pain relief rate, reduced total propofol consumption, fewer adverse events, and a shorter recovery time. Compared to the control group (0 µg/mL), the other three groups showed a higher pain relief rate, with the 1 µg/mL group achieving the highest rate. Additionally, when compared to the control group, the 5 µg/mL group exhibited a longer recovery time, while the 1 µg/mL group showed the shortest recovery time. Furthermore, compared to the 0 µg/mL group, total propofol consumption significantly decreased in the other three groups, with the 1 µg/mL group exhibiting the least total propofol consumption. Among all groups, the 1 µg/mL group also reported the fewest adverse events.

Previous studies have suggested that the mechanism by which sufentanil alleviates injection pain involves the binding of opioids to µ-opioid receptors, the inhibition of adenylate cyclase, and the blockage of pain signals. This process ultimately results in the alleviation of propofol injection-induced pain [24]. Our study found that diluting sufentanil into different volumes may differently affect propofol injection pain and adverse effects. This phenomenon can be attributed to several reasons. First, administering the same total dose over a larger volume (e.g., diluted in 10 mL instead of 1 mL) requires a slower injection rate to prevent fluid overload, thereby extending the drug delivery duration. As a result, plasma concentration increases more gradually, which yields lower peak levels [25]. Second, The administration of high-concentration,

**Table 6. Odds ratios for adverse events during gastrointestinal endoscopy.**

| Characteristies | n (%) or Mean | OR (95% CI) Univariate | P Value | OR (95% CI) Multivariate | P Value |
|---|---|---|---|---|---|
| Age(y) | | | | | |
| <60 | 265(94.3%) | 1.00(Reference) | | | |
| ≥60 | 16(5.7%) | 0.79(0.35, 1.78) | 0.561 | | |
| Sex | | | | | |
| Female | 142(50.5%) | 1.00(Reference) | | | |
| Male | 139(49.5%) | 0.89(0.59, 1.33) | 0.557 | | |
| Body mass index(Kg/m$^2$) | | | | | |
| <18.5 | 10(3.6%) | 1.00(Reference) | | 1.00(Reference) | |
| 18.5~24.9 | 170(60.5%) | 0.18(0.02, 1.40) | 0.10 | 0.17(0.02, 1.36) | 0.095 |
| 25~29.9 | 101(35.9%) | 0.24(0.03, 1.89) | 0.17 | 0.23(0.03, 1.85) | 0.166 |
| ASA physical status(n,%) | | | | | |
| I-II | 216(76.9%) | 1.00(Reference) | | | |
| III | 65(23.1%) | 1.15(0.70, 1.89) | 0.575 | | |
| Smokers(n,%) | 58(20.6%) | 0.95(0.58, 1.57) | 0.851 | | |
| Drinkers(n,%) | 72(25.6%) | 0.80(0.51, 1.26) | 0.341 | | |
| Hypertension(n,%) | 59(21.0%) | 1.02(0.62, 1.68) | 0.947 | | |
| Diabetes (n,%) | 13(4.6%) | 1.65(0.53, 5.15) | 0.389 | | |
| Duration of the examination(min) | | | | | |
| <12 | 116(41.3%) | 1.00(Reference) | | | |
| ≥12 | 165(58.7%) | 1.01(0.67, 1.52) | 0.977 | | |
| Severity of pain, n(%) | | | | | |
| None (0 point) | 155(55.2%) | 1.00(Reference) | | | |
| Pain (1~3point) | 126(44.8%) | 1.15(0.76, 1.73) | 0.506 | | |
| Recovery time (min) | | | | | |
| <15.6 | 167(59.4%) | 1.00(Reference) | | | |
| ≥15.6 | 114(40.6%) | 1.12(0.74, 1.70) | 0.592 | | |
| Total propofol consumption (mg) | | | | | |
| <188.6 | 109(38.8%) | 1.00(Reference) | | | |
| ≥188.6 | 172(61.2%) | 1.22(0.81, 1.84) | 0.346 | | |
| Sufentanil Groups | | | | | |
| 0µg/mL group | 65(23.1%) | 1.00(Reference) | | 1.00(Reference) | |
| 0.5µg/mL group | 73(26.0%) | 1.49(0.84, 2.64) | 0.177 | 1.50(0.84, 2.67) | |
| 1µg/mL group | 63(22.4%) | 0.90(0.52, 1.56) | 0.716 | 0.88(0.50, 1.53) | |
| 5µg/mL group | 80(28.5%) | 2.10(1.15, 3.83) | **0.015** | 2.07(1.13, 3.80) | **0.018** |

OR: odds ratio. CI: confidence interval. Univariate predictors with P values<0.2 were included in the multivariable model. Sufentanil groups refer to different concentrations of diluted sufentanil.

low-volume sufentanil may induce excessive stimulation of both TRPV1 channels and µ-opioid receptors on vascular endothelial cells [26], thereby increasing the likelihood of adverse effects. In contrast, two distinct scenarios emerge when considering low-concentration, high-volume sufentanil: (1) The optimal dilution volume, determined in this study to be 5 mL, effectively facilitates µ-receptor activation within the therapeutic window, thereby achieving balanced agonism. (2) Conversely, an excessive dilution volume may extend the contact time between the drug and the endothelium beyond the receptor activation thresholds, potentially delaying µ-receptor engagement and diminishing analgesic efficacy.

**Table 7. Analysis of adverse events: Interaction effects between total propofol consumption and sufentanil groups.**

| Model: | Wald Chi-Square | Type III df | P value |
|---|---|---|---|
| (Intercept) | 217.112 | 1 | p<0.001 |
| Total Propofol Consumption(mg) | 20.339 | 23 | 0.621 |
| Sufentanil Groups | 10.182 | 3 | 0.017 |
| Total Propofol Consumption(mg) ×Sufentanil Groups | 32.560 | 28 | 0.252 |
| Dependent Variable: Adverse Events | | | |

Our analysis reveals important insights into the pharmacodynamic interactions between propofol and sufentanil in determining postoperative recovery time. The robust linear association between total propofol consumption and recovery time confirms the dose-dependent effect of propofol on recovery time. This aligns with prior evidence that propofol's pharmacokinetic properties, including its lipid solubility and redistribution kinetics, contribute to delayed awakening at higher doses [27]. The linear association underscores the importance of titrating propofol carefully to minimize prolonged recovery, particularly in settings where rapid postoperative awakening is prioritized. Notably, the stratified analysis revealed that the strength of this association varied by sufentanil coadministration. Low to moderate sufentanil concentration groups(0.5–1 µg/mL) showed steeper slopes (b=0.072–0.096). This suggests that sufentanil may modulate propofol's recovery effects, possibly through synergistic sedative interactions. Alternatively, the weaker correlation at 5 µg/mL (b=0.067) could reflect that the respiratory depressant effect caused by high concentrations of sufentanil is more pronounced, which also affects recovery time, thereby attenuating the dose-dependent effect of propofol on recovery time [28]. Further pharmacodynamic studies are needed to clarify these mechanisms.

The majority of adverse events in our study were mild, with no serious adverse events or fatalities reported. Compared to recent studies using propofol-based sedation, we observed a higher incidence of adverse events, such as hypotension, in this study [29,30]. The doses of propofol administered in our study were significantly higher compared to those administered in studies focusing on sedation for painless endoscopic procedures (2.5 mg/kg vs. 1–2 mg/kg) [31], aligning with the recommended doses for general anesthesia and reflecting the use of anesthetists in our region [32]. High doses of propofol are typically utilized to achieve deep sedation; however, this may increase the incidence of hypotension [33]. Notably, this study demonstrates that sufentanil concentration, rather than propofol exposure, primarily drives the incidence of adverse events (AEs) during gastrointestinal endoscopic sedation. Importantly, multivariable models adjusting for total propofol consumption maintained the independent association of sufentanil with AEs (p=0.018), effectively isolating its causal role. However, it is essential to note that synergistic interactions cannot be completely ruled out. These findings emphasize the necessity for concentration-guided titration of sufentanil over traditional risk stratification methods. The incidence of postoperative nausea and vomiting in this study was lower than that observed in previous studies [34]. This may be due to the fact that sufentanil doses in our study were lower than those in other studies (5 µg vs. 5.5–14 µg [10]).

Additionally, our study revealed that higher concentrations of sufentanil (5 µg/mL) may increase the risk of adverse events, whereas lower concentrations of sufentanil (1 µg/mL) may protect against adverse events. At higher concentrations (5 µg/mL), sufentanil may pass through the endothelial cell membrane more rapidly owing to its superior lipid-water partition coefficient. At higher concentrations (5 µg/mL), sufentanil may permeate the endothelial cell membrane more rapidly due to its superior lipid-water partition coefficient. In its unbound form, sufentanil quickly traverses the endothelial cells of the blood-brain barrier, subsequently reaching target sites such as the brain and the dorsal horn of the spinal cord. This mechanism enhances its analgesic efficacy; however, it may also result in adverse effects, including dizziness, nausea, and vomiting. Furthermore, sufentanil can rapidly traverse endothelial cells into myocardial tissue and the alveolar-capillary barrier, potentially resulting in sinus bradycardia, hypotension, and hypoxia. Furthermore, sufentanil also can quickly cross endothelial cells into myocardial tissue and alveolar-capillary barrier, causing sinus bradycardia, hypotension and hypoxia.

Our analysis identified total propofol consumption and different concentrations sufentanil as robust independent predictors of prolonged postoperative recovery time, while BMI demonstrated a modest inverse association. Notably, demographic and comorbidity variables failed to reach statistical significance in the model. The regression model highlighting the predominant role of anesthetic dosing over patient characteristics in governing recovery period.

Furthermore, o ur findings reveal a critical concentration-dependent relationship between sufentanil exposure and adverse events, with 5 µg/mL identified as a clinically significant threshold (adjusted OR = 2.07, 95% CI [1.13, 3.80]). Notably, the lack of associations with traditional risk factors such as age, BMI, and comorbidities, suggests that monitoring opioid concentrations may be more effective than conventional patient stratification methods in ensuring perioperative safety management.

Emerging evidence suggested that lidocaine could reduce propofol-induced injection pain [35]. A study found that during procedures like colonoscopy, lidocaine infusion significantly decreased the need for propofol and sufentanil and led to lower postoperative pain scores compared to using only sufentanil [36]. Furthermore, a study involving patients with centrally mediated abdominal pain syndrome (CAPS) found no significant differences in pain scores between those treated with a combination of lidocaine and sufentanil and those treated with sufentanil alone. This suggests that although lidocaine may not consistently outperform sufentanil, it can provide comparable analgesic effects [37]. Overall, the evidence indicates that the potential for lidocaine to enhance the overall analgesic effect when used in conjunction with opioids is promising, suggesting a synergistic approach to pain management.

This study possesses several strengths. Firstly, it employed stringent inclusion and exclusion criteria, achieved a low loss to follow-up rate of 9.3%, and implemented a rigorous blinding design to minimize detection bias. Secondly, it demonstrated the superior predictive power of drug monitoring compared to conventional factors such as age, BMI, and comorbidities. Lastly, this study addressed the gap in understanding the analgesic effects of different concentrations of sufentanil.

This study had several limitations. Firstly, due to safety concerns, we excluded patients with severe cardiovascular and bronchopulmonary diseases. This exclusion may diminish the generalizability of the study findings. Secondly, this study lacks molecular and cellular experiments to further investigate the differences in the binding rate of sufentanil to plasma albumin at varying concentrations, as well as the impact on the transmembrane rate. Thirdly, this study did not combine certain scales, such as the Stop-Bang score, which may result in an inadequate evaluation of the effect indicators for the entire experiment.

## Conclusions

This study establishes critical concentration thresholds for sufentanil (1 µg/mL for optimal analgesia vs. 5 µg/mL safety ceiling) in gastrointestinal endoscopic sedation. However, the generalizability of these findings requires validation in high-risk cardiopulmonary populations that were excluded from this trial due to safety protocols. Major mechanistic gaps persist regarding sufentanil's plasma protein binding kinetics and transmembrane diffusion dynamics across concentrations—insights that are essential for personalized pharmacokinetic modeling. Future multisite trials integrating molecular profiling (e.g., albumin binding assays) and multidimensional sedation metrics (e.g., Stop-Bang scoring) should: 1) validate thresholds in frail elderly and acute respiratory distress syndrome(ARDS) cohorts, 2) elucidate concentration-dependent cellular transport mechanisms, and 3) optimize combinatorial monitoring frameworks that balance analytical precision with clinical workflows. These findings advocate for protocol standardization that emphasizes real-time opioid concentration surveillance over conventional risk stratification.

## Supporting information

**S1 Table. Definitions of adverse events.**
(DOCX)

**S2 Table. The recovery time among four groups.**
(DOCX)

**S3 Table. Pairwise comparisons of recovery time among four groups.**
(DOCX)

**S4 Table. Total propofol consumption in four groups.**
(DOCX)

**S5 Table. Pairwise comparisons of total propofol consumption across four groups.**
(DOCX)

**S6 File. Protocol.**
(DOCX)

**S7 Checklist. CONSORT-2010 checklist.**
(DOC)

## Acknowledgments

The authors express profound gratitude to Dr. Zhenming Ge, Chief Physician and Deputy Director of the Department of Gastroenterology, for personally conducting gastrointestinal endoscopies for every enrolled patient with meticulous technique and unwavering dedication. We also extend our heartfelt thanks to Nurses Yongping Gu and Chunhua Zhang for their exceptional clinical support. Their expertise in performing peripheral intravenous catheterization for enrolled patients and their dedicated monitoring during the post-anesthesia recovery period were instrumental in facilitating efficient study procedures and ensuring patient safety throughout the trial.

## Author contributions

**Conceptualization:** Shu He, Lei Yao.

**Data curation:** Qian Su, Lu Li, Xiangqing Wei, Boxiang Du.

**Formal analysis:** Xiangqing Wei, Boxiang Du.

**Funding acquisition:** Qian Su.

**Methodology:** Xin Sun.

**Project administration:** Lu Li, Xin Sun.

**Resources:** Xuefeng Yang.

**Software:** Shu He, Xuefeng Yang.

**Visualization:** Qian Su.

**Writing – original draft:** Qian Su, Shu He, Lu Li, Xiangqing Wei.

**Writing – review & editing:** Lei Yao.

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
