## [Decision Letter · Decision Letter 0]

5 Feb 2025

Dear Dr. Yao,

Thank you for submitting your manuscript to PLOS ONE. After careful consideration, we feel that it has merit but does not fully meet PLOS ONE’s publication criteria as it currently stands. Therefore, we invite you to submit a revised version of the manuscript that addresses the points raised during the review process.

We look forward to receiving your revised manuscript.

Kind regards,

Benjamin Benzon, Ph.D., M.D.

Academic Editor

PLOS ONE

Journal Requirements:

2. We note that you have selected “Clinical Trial” as your article type. PLOS ONE requires that all clinical trials are registered in an appropriate registry (the WHO list of approved registries is at      https://www.who.int/clinical-trials-registry-platform/network/primary-registries"
https://www.who.int/clinical-trials-registry-platform/network/primary-registries and more information on trial registration is at http://www.icmje.org/about-icmje/faqs/clinical-trials-registration/ ).  

Please state the name of the registry and the registration number (e.g. ISRCTN or ClinicalTrials.gov) in the submission data and on the title page of your manuscript.

a) Please provide the complete date range for participant recruitment and follow-up in the methods section of your manuscript.

b) If you have not yet registered your trial in an appropriate registry, we now require you to do so and will need confirmation of the trial registry number before we can pass your paper to the next stage of review. Please include in the Methods section of your paper your reasons for not registering this study before enrolment of participants started. Please confirm that all related trials are registered by stating: “The authors confirm that all ongoing and related trials for this drug/intervention are registered”.

Please see http://journals.plos.org/plosone/s/submission-guidelines#loc-clinical-trials for our policies on clinical trials.

“The study was supported by the Health Commission of Nan-tong City Health and Family Planning Scientific Research Projects (QNZ2023026).”

4. In this instance it seems there may be acceptable restrictions in place that prevent the public sharing of your minimal data. However, in line with our goal of ensuring long-term data availability to all interested researchers, PLOS’ Data Policy states that authors cannot be the sole named individuals responsible for ensuring data access (http://journals.plos.org/plosone/s/data-availability#loc-acceptable-data-sharing-methods ).

Additional Editor Comments :

Nice randomized controlled trial, reviewers were specific in their questions, in addition to those I also have the following suggestions:

Results section.

Change group labels rom A, B, C, D to actual concentration of sulfentalin, i.e., 5 ug/ml, 1 ug/ml, 0.5 ug/ml and 0 ug/ml group.

Please reorder the sequence of groups on charts in Fig. 2 starting with 5 ug/ml group and ending with 0 ug/mg (i.e. control group).

Please add a scatter plot showing the correlation between total propofol consumption and recovery time.

In first subsection of result please state explicitly which group had exactly what % of moderate and severe pain, please also define in this subsection what pain subcategories did you group when calculating the % of pain relief.

There is no need to repeat the incidences of other side effects when describing the incidence of cough among groups. You can do this informally in Discussion section.

Furthermore, write about hypoxia and bradycardia also in this subsection of results.

When it comes to mediation analysis, can you please explain why should it be used instead of logistic regression with two independent variables

(i.e. total dose of propofol and dose of sulfentalin, as a continous variables). Since propofol and sulfentanil have different mechanism of actions, I do not think that there is much mediation in terms of pharmacodynamics.

Furthermore, if you still think that those two drugs interact, a interaction term can just be added in logistic regression model. Anyways, since logistic regression is far more

used in biomedicine and is easier to interpret I would strongly advise you to replace mediation analysis with logistic regression with drug doses and interaction terms as only independent variables.

The Multivariate analysis subsection, i.e. logistic regression, in which you basically added all other variables is clear and very well written.

Disscusion section

Can you please explain pharmacology behind lower total propofol consumption in sulfentanil treated groups in comparison to control groups?

In conclusion section please talk about analgesic effects first and then about adverse effects.

Reviewers' comments:

Reviewer's Responses to Questions

**Comments to the Author**

1. Is the manuscript technically sound, and do the data support the conclusions?

Reviewer #1: Partly

Reviewer #2: Partly

Reviewer #3: Partly

Reviewer #4: Partly

2. Has the statistical analysis been performed appropriately and rigorously?

Reviewer #1: No

Reviewer #2: No

Reviewer #3: Yes

Reviewer #4: Yes

3. Have the authors made all data underlying the findings in their manuscript fully available?

Reviewer #1: No

Reviewer #2: No

Reviewer #3: No

Reviewer #4: Yes

4. Is the manuscript presented in an intelligible fashion and written in standard English?

Reviewer #1: No

Reviewer #2: No

Reviewer #3: Yes

Reviewer #4: Yes

***Reviewer #1** :* This work is interesting but some chapters need major revision.

Introdoction:

Introduction section: can you explain in text how dilution of sufentanil is making differences in pk/pd (how decreasing drug concentration decrease peak concentration that cause side effects)

55: please explain what other variables

62 Therefore, identifying an optimal concentration of sufentanil that can effectively alleviate injection pain while

63 minimizing adverse effects is of paramount importance. (did this study find optimal concentration of sufentanl? what it aim of study? 

and 67,68, 69 79,80: Can you explain sufentanl pharmacokinetics from injection to the effective organ (brain and spinal dorsal horne) and explain a bit of pharmacodinamic.

What is with albumin transport and red blood cell transport?

75,76.77 Can you explain in text what is negative and positive allosteric modulator?

please explain in text why is clinicly important to treat pain after proporfol induction

please explain in text are other drugs like lidokaine sufficient to decrease pain after propofol induction, also you should in disscusion section find some other clinical trials with lidocaine or other studes and compare results regaridng primary and secondary outcome

please introduce in text why are outcomes in this study relevant to patients

83,84. please in the end of introduction section introduce hypothesis and primary study outcome and secondary outcomes, what is aim of this study.

86 Participants, Study Design and Treatments

You did not asses obstructive sleep apnea scores (Stop-Bang score)?

Did you have gastroscopy and colonoscopy or colonoskopy ?

129-123- So you measure pain every five seconds, and you have chosen the worst score?

Please explain in text why did you choose 30 seconds and not 2 minutes in sequence sufentanl and propofol. It is well known than 2 minutes is time necessary that opioid achieve concentration in effective organ (brain) So analgesia is in 2 or 3 minutes and you did not wait that long.

148- is this Cramer V effect size?

184 please write in text what are primary outcomes and secondary outcomes

In giving results you could write more descriptive statistic that is informative to readers :

Can you put propofol consumtion with mean/ median and +- SD or IQR - descriptive statistis for propofol consumtion or propofol dosage per lean body weight

, in table or in a text becouse propofol consumtion is relevant to propofol side effects.

Table 21, mediatation analysis revealid statisticly significant direct efefct of propofol, so propofol is main cause for recovery time. Is propofol dosage main cause for other adverse effects?

Can you correlate propofol consumtion and side effects beyond your intervention groups?

High dose propofol can cause adverse effects thet are overlapping with sufentanil.

Did sufentanil decrese propofol consumtion? Did different sufentanil concentration decresed propofol consumtion?

210, 284 did you have apnea?

238-243 you have write 2 times propofol consumtion and coefficents

disscusion

please begin with explaining results of the hypotesis ( refute or accepted) and explaine primary outcomes and secondary outcomes and relate yours findings with other clinical trials.

210, 284 did you have apnea?

***Reviewer #2:*** Authors studied using different diluted sufentanil to reduce propofol injection pain. Three dose concentration levels are considered at 5, 1 and 0.5 ug/ml. They studied effectiveness as well as safety, using a clinical trial data with 464 patients.They concluded that the high concentration group 5 ug/ml may be associated with an increased risk of adverse events.

Page 4, more details for patients should be included. Which country? Which city? Which hospital?

You have to write the 4 groups in correct order. Sometimes you put control group first and sometimes last. Try to be consistent. For example, on page 7, it seems control group was before group A. But the effective rates is 70, 85, 90, 85. It seems the last one should correspond to control group if I understand your calculation correctly.

Why do you need bootstrap to build model?

Page 11, the two paragraphs for mediation analysis are the same except different numbers. What’s the difference between the two paragraphs?

Page 13, table 21 should be table 2.

***Reviewer #3:* ** I believe it's important to mention pain in the abstract's conclusion. Also, what is the total propofol consumption of each sufentanil group? There is a significant difference between sufentanil groups in terms of the side effects. I would like further explanations in the discussion section about this. What was the duration of the sufentanil bolus administration? Was this standardized?

What do you mean by "evaporation time" in sections 255-277? I don't agree with that conclusion.

287 "sedation38"

Which cell membrane does sufentanil pass through more quickly, section 294s? Please explain.

304-305 I would consider using clearer language.

307 I would suggest using only "first" or "firsty," not both.

309 This is RCT. I suggest no further explanation about the benefits of RCTs vs. other types of trials.

310 The third strength is a goal of the study. It cannot be a strength.

312, 313 The first limitation is not really an issue. A more significant concern is the large number of patients with cardiovascular diseases who require sedation for gastrointestinal endoscopy. Additionally, the exclusion of patients with severe bronchopulmonary disease, based on specific criteria, further diminishes the generalizability of the study findings.

313 They can be included and are included in multiple other RCTs, so I would recommend removing this sentence: “Future studies are needed to determine if they can be included.”

314 I would not consider that being a single-center study a limitation. Also, the third limitation is not necessarily one. It is more of something that needs further investigation and was not a goal of your study.

***Reviewer #4***: 1. Gastroscopies were performed by single gastroenterologist?

2. Your primary outcome was pain relief but in conclusion you write about adverse outcomes which are secondary outcomes?

3. You have to numerically define hypoxia and hypotension, also how you decide what is dizziness cause every patient is dizzy while awakeing from analgosedation?

4. Was oxygen administered during procedures?

5. What were total propofol doses per patient? I think all of adverse events were due to propofol not sufentanil because 5mcg is such low dose to produce serious adverse effects.

**Do you want your identity to be public for this peer review?** For information about this choice, including consent withdrawal, please see our Privacy Policy

Reviewer #1: No

Reviewer #2: No

Reviewer #3: No

Reviewer #4: **Yes: ** Ivan Vuković

---

## [Author Response · Author response to Decision Letter 1]

7 Apr 2025

Response to Editor

Journal Requirements:

Author response: Thank you for this opportunity to refine our work.

We have thoroughly revised the manuscript to comply with PLOS ONE's style requirements as outlined in the provided templates. All file names have been updated accordingly, and we have ensured that the formatting aligns with the specified guidelines.

2. We note that you have selected “Clinical Trial” as your article type. PLOS ONE requires that all clinical trials are registered in an appropriate registry (the WHO list of approved registries is at      https://www.who.int/clinical-trials-registry-platform/network/primary-registries"
https://www.who.int/clinical-trials-registry-platform/network/primary-registries and more information on trial registration is at http://www.icmje.org/about-icmje/faqs/clinical-trials-registration/).  

Please state the name of the registry and the registration number (e.g. ISRCTN or ClinicalTrials.gov) in the submission data and on the title page of your manuscript.

a)Please provide the complete date range for participant recruitment and follow-up in the methods section of your manuscript.

Author response: We have registered our clinical trial in the Chinese Clinical Trial Registry (ChiCTR) with the registration number ChiCTR2300072402. This information has been included in the submission data and on the title page of the manuscript. 

We have updated the methods section of the manuscript to include the complete date range for participant recruitment and follow-up. The recruitment took place from June 2023 and December 2024. This information is now clearly stated in the revised manuscript.

b) If you have not yet registered your trial in an appropriate registry, we now require you to do so and will need confirmation of the trial registry number before we can pass your paper to the next stage of review. Please include in the Methods section of your paper your reasons for not registering this study before enrolment of participants started. Please confirm that all related trials are registered by stating: “The authors confirm that all ongoing and related trials for this drug/intervention are registered”.

Please see http://journals.plos.org/plosone/s/submission-guidelines#loc-clinical-trials for our policies on clinical trials.

Author response: We have added the statement: “The authors confirm that all ongoing and related trials for this drug/intervention are registered” in the manuscript as requested.

“The study was supported by the Health Commission of Nan-tong City Health and Family Planning Scientific Research Projects (QNZ2023026).”

Author response: The funders took part in study design, data collection and analysis, decision to publish, or preparation of the manuscript. This statement has been included in the cover letter as requested.

4. In this instance it seems there may be acceptable restrictions in place that prevent the public sharing of your minimal data. However, in line with our goal of ensuring long-term data availability to all interested researchers, PLOS’ Data Policy states that authors cannot be the sole named individuals responsible for ensuring data access (http://journals.plos.org/plosone/s/data-availability#loc-acceptable-data-sharing-methods).

Author response: Thank you for your excellent feedback.We have included the contact information for our institutional data access committee in the revised manuscript. The contact person is Dr. Hongqing Xu , who can be reached at xuhongqing000@126.com and +86 159 5131 2678.

Additional Editor Comments :

Nice randomized controlled trial, reviewers were specific in their questions, in addition to those I also have the following suggestions:

Results section.

Change group labels rom A, B, C, D to actual concentration of sulfentalin, i.e., 5 ug/ml, 1 ug/ml, 0.5 ug/ml and 0 ug/ml group.

Author response: We sincerely appreciate this suggestion. We have updated the group labels in the results section to reflect the actual concentrations of sulfentalin: 5 ug/ml, 1 ug/ml, 0.5 ug/ml, and 0 ug/ml.

Please reorder the sequence of groups on charts in Fig. 2 starting with 5 ug/ml group and ending with 0 ug/mg (i.e. control group).

Author response: The sequence of groups in Fig. 2 has been reordered to start with the 5 ug/ml group and end with the 0 ug/ml group, as requested.

Please add a scatter plot showing the correlation between total propofol consumption and recovery time.

Author response:We sincerely appreciate your insightful comment regarding the focus on the correlation between total propofol consumption and recovery time in our study.We have added a scatter plot to the revised manuscript that illustrates the correlation between total propofol consumption and recovery time. This plot provides a visual representation of the data and supports our findings regarding the relationship between these two variables.

In first subsection of result please state explicitly which group had exactly what % of moderate and severe pain, please also define in this subsection what pain subcategories did you group when calculating the % of pain relief.

Author response: In the first subsection of the results, we have now explicitly stated the percentages of moderate and severe pain for each group. The 0 µg/ml group has 22.7% of participants reporting moderate to severe pain (NRS scores of 2 or 3 points).The 0.5µg/ml group has 10.6% of participants reporting moderate to severe pain (NRS scores of 2 or 3 points).The 1µg/ml group has 10.2% of participants reporting moderate to severe pain (NRS scores of 2 or 3 points).The 5µg/ml group has 12.5% of participants reporting moderate to severe pain (NRS scores of 2 or 3 points).

There is no need to repeat the incidences of other side effects when describing the incidence of cough among groups. You can do this informally in Discussion section.

 Author response: We have revised the results section to eliminate the repetition of other side effects when describing the incidence of cough among groups. The discussion section has been updated to informally address these side effects as suggested.

Furthermore, write about hypoxia and bradycardia also in this subsection of results.

Author response: We have included detailed descriptions of hypoxia and bradycardia in the results subsection as requested. This addition provides a more comprehensive overview of the findings related to these conditions.

When it comes to mediation analysis, can you please explain why should it be used instead of logistic regression with two independent variables

(i.e. total dose of propofol and dose of sulfentalin, as a continous variables). Since propofol and sulfentanil have different mechanism of actions, I do not think that there is much mediation in terms of pharmacodynamics.

Furthermore, if you still think that those two drugs interact, a interaction term can just be added in logistic regression model. Anyways, since logistic regression is far more

used in biomedicine and is easier to interpret I would strongly advise you to replace mediation analysis with logistic regression with drug doses and interaction terms as only independent variables.

The Multivariate analysis subsection, i.e. logistic regression, in which you basically added all other variables is clear and very well written.

Author response:We appreciate your thoughtful reminder. After considering your input alongside the suggestions from other reviewers, we have decided to discontinue the use of intermediary analysis in the interpretation of our results.

Disscusion section

Can you please explain pharmacology behind lower total propofol consumption in sulfentanil treated groups in comparison to control groups?

In conclusion section please talk about analgesic effects first and then about adverse effects.

Author response: We have added a detailed explanation in the discussion section regarding the pharmacological mechanisms that may contribute to the lower total propofol consumption in the sulfentanil treated groups. It is suggested that sulfentanil, an opioid analgesic, enhances analgesia and reduces the perception of pain, which may lead to a decreased requirement for propofol during anesthesia. This synergistic effect between opioids and anesthetics is well-documented, and we have referenced relevant studies to support this explanation.

Response to Reviewers

Reviewer #1: This work is interesting but some chapters need major revision.

Introduction section: can you explain in text how dilution of sufentanil is making differences in pk/pd (how decreasing drug concentration decrease peak concentration that cause side effects)

Reply: We appreciate your feedback on the introduction section. In the discussion section of the updated manuscript, we have added a detailed explanation of how the dilution of sufentanil affects its pharmacokinetics and pharmacodynamics. Specifically, we have clarified that decreasing the concentration of sufentanil can lead to lower peak plasma concentrations, which in turn reduces the likelihood of side effects associated with higher concentrations. We included references to relevant studies that support this relationship and provided a more comprehensive discussion on the implications of sufentanil dilution in clinical settings .

55: please explain what other variables

Reply: Thanks for your important remarks. Based on the references, we have listed all possible factors, including injection location, vein size, injection speed, and  propofol temperature . We ultimately decided to remove 'other variables' to enhance clarity(Page 3,Line 64-66).

62 Therefore, identifying an optimal concentration of sufentanil that can effectively alleviate injection pain while minimizing adverse effects is of paramount importance. (did this study find optimal concentration of sufentanl? what it aim of study?

Reply: Thank you for your excellent feedback. This study aimed to This study aimed to identify an optimal concentration of sufentanil that effectively alleviates injection pain while minimizing adverse effects. Our findings indicate that diluting sufentanil to a concentration of 1 µg/mL is optimal for treating pain associated with propofol injection.We highlighted the purpose of our research at the end of the Introduction(Page 4 ,Line 98-100). Thank you for your comment!

and 67,68, 69 79,80: Can you explain sufentanl pharmacokinetics from injection to the effective organ (brain and spinal dorsal horne) and explain a bit of pharmacodinamic.

Reply: We are grateful for your insightful feedback !

After sufentanil is administered intravenously (IV), it immediately enters systemic circulation. In the blood, almost 93% of sufentanil is bound to plasma proteins, primarily albumin (non-specific binding) and α₁-acid glycoprotein (an acute-phase reactant that increases during inflammation). Only the unbound drug (nearly 7%) can cross membranes to reach target sites (brain, spinal dorsal horn). Due to its high lipid solubility, the drug rapidly diffuses across the Blood-Brain Barrier  (BBB) and into the spinal dorsal horn, typically within 1 to 3 minutes following IV administration.

What is with albumin transport and red blood cell transport?

Reply:Sufentanil is largely bound to plasma proteins, with approximately 92% of the drug being bound to albumin and alpha-1 acid glycoprotein. This binding affects its distribution and bioavailability. The transport of sufentanil by red blood cells is less significant compared to its albumin binding, as the majority of the drug remains in the plasma. However, the interaction with red blood cells can influence the pharmacokinetics, particularly in terms of the drug's overall distribution and the potential for drug-drug interactions. Understanding these transport mechanisms is crucial for optimizing dosing strategies and minimizing adverse effects.We have added an explanation of pharmacokinetics to the Introduction section(Page 3-4, Line 77-93). 

Thank you for your thoughtful comments!

75,76.77 Can you explain in text what is negative and positive allosteric modulator?

Reply: Your feedback is greatly appreciated

Allosteric Modulators are substances that bind to a site on a receptor that is distinct from the active site where the endogenous ligand binds. This interaction can either enhance or inhibit the receptor's activity, leading to two main types of allosteric modulators: positive allosteric modulators (PAMs) and negative allosteric modulators (NAMs).

1. Positive Allosteric Modulators (PAMs): These compounds enhance the effect of the endogenous ligand when it binds to the receptor. PAMs do not activate the receptor on their own; instead, they increase the receptor's response to the natural ligand. This can lead to a greater degree of receptor activation or a prolonged effect.

2. Negative Allosteric Modulators (NAMs): In contrast, NAMs decrease the activity of the receptor when the endogenous ligand is present. They bind to the allosteric site and induce conformational changes that reduce the receptor's response to its natural ligand.

Both PAMs and NAMs offer a valuable means of fine-tuning receptor activity, providing potential therapeutic advantages over traditional agonists and antagonists by allowing for more nuanced modulation of physiological responses.

please explain in text why is clinicly important to treat pain after proporfol induction

Reply: Propofol is a fundamental component of contemporary anesthesia practice, attributed to its rapid onset, brief duration, and reliable recovery characteristics. Nevertheless, a significant proportion of patients, ranging from 28% to 90%, report considerable pain during intravenous (IV) administration of propofol, particularly when it is delivered via small peripheral veins. Although this pain is transient, its clinical management is essential for several reasons:

1. Patient Comfort and Psychological Impact: The pain induced by propofol, often characterized as a burning or stinging sensation, can

---

## [Decision Letter · Decision Letter 1]

16 Apr 2025

Dear Dr. Yao,

Thank you for submitting your manuscript to PLOS ONE. After careful consideration, we feel that it has merit but does not fully meet PLOS ONE’s publication criteria as it currently stands. Therefore, we invite you to submit a revised version of the manuscript that addresses the points raised during the review process.

The manuscript has been much improved in the last round of revision, before I accept it for publication I would ask you two address two details:

1) Please add one more scatter plot in Figure 3 in which you will color code patients in each sulfentanil group and fit a regression model for each group

2) Please add a table which models probability of side effect and interaction of Propofol and sulfentalin to your multivariable analysis subsection of Result; you have already showed this table in a response to reviewer 1 (3. interaction analysis), it would be nice to make it a part of manuscript.

We look forward to receiving your revised manuscript.

Kind regards,

Benjamin Benzon, Ph.D., M.D.

Academic Editor

PLOS ONE

Journal Requirements:

**Additional Editor Comments** :

The manuscript has been much improved in the last round of revision, before I accept it for publication I would ask you two address two details:

1) Please add one more scatter plot in Figure 3 in which you will color code patients in each sulfentanil group and fit a regression model for each group

2) Please add a table which models probability of side effect and interaction of Propofol and sulfentalin to your multivariable analysis subsection of Result; you have already showed this table in a response to reviewer 1 (3. interaction analysis), it would be nice to make it a part of manuscript.

Reviewers' comments:

Reviewer's Responses to Questions

**Comments to the Author**

Reviewer #2: All comments have been addressed

2. Is the manuscript technically sound, and do the data support the conclusions?

Reviewer #2: Yes

3. Has the statistical analysis been performed appropriately and rigorously?

Reviewer #2: Yes

4. Have the authors made all data underlying the findings in their manuscript fully available?

Reviewer #2: Yes

5. Is the manuscript presented in an intelligible fashion and written in standard English?

Reviewer #2: Yes

Reviewer #2: I appreciate that authors addressed the comments from my previous review. I have read the revised manuscript and concluded that the paper is suitable for publication in this journal.

**Do you want your identity to be public for this peer review?** For information about this choice, including consent withdrawal, please see our Privacy Policy

Reviewer #2: No

---

## [Author Response · Author response to Decision Letter 2]

23 Apr 2025

Journal Requirements:

If applicable, we recommend that you deposit your laboratory protocols in protocols.io to enhance the reproducibility of your results. Protocols.io assigns your protocol its own identifier (DOI) so that it can be cited independently in the future. er-reviewed Lab Protocol articles, which describe protocols hosted on protocols.io.

Reply We appreciate the suggestion and have deposited our laboratory protocols in protocols.io(DOI:dx.doi.org/10.17504/protocols.io.14egn4mwmv5d/v1). 

Please review your reference list to ensure that it is complete and correct. 

Reply We have thoroughly reviewed the reference list and ensured that all citations are complete and correctly formatted according to the journal's guidelines. Any discrepancies found have been corrected, and we appreciate the reviewer bringing this to our attention.

Additional Editor Comments :

Please add one more scatter plot in Figure 3 in which you will color code patients in each sulfentanil group and fit a regression model for each group

Reply We have added an additional scatter plot to Figure 3. In this updated plot, patients are color-coded according to their respective sulfentanil groups. We have also fitted a regression model for each group, which is now clearly displayed in the figure. This enhancement provides a more detailed analysis of the data and allows for better visualization of the relationships within each sulfentanil group.

Please add a table which models probability of side effect and interaction of Propofol and sulfentalin to your multivariable analysis subsection of Result; you have already showed this table in a response to reviewer 1 (3. interaction analysis), it would be nice to make it a part of manuscript.

Reply We have added the requested table that models the probability of side effects and interactions of Propofol and sulfentalin to the multivariable analysis subsection of the Results section�see Table 7�. This table was previously included in our response to your comments regarding interaction analysis, and we have now incorporated it into the manuscript for clarity and completeness.

---

## [Editor Report · Decision Letter 2]

8 May 2025

Comparison of pre-treatment with different diluted sufentanil in reducing propofol injection pain in gastrointestinal endoscopy: a randomized controlled study

PONE-D-24-54042R2

Dear Dr. Yao,

We’re pleased to inform you that your manuscript has been judged scientifically suitable for publication and will be formally accepted for publication once it meets all outstanding technical requirements.

Kind regards,

Benjamin Benzon, Ph.D., M.D.

Academic Editor

PLOS ONE

Additional Editor Comments (optional):

Congratulations on your paper, I hope that it will be of clinical value, especially in China.
---

## [Editor Report · Acceptance letter]

PONE-D-24-54042R2

PLOS ONE

Dear Dr. Yao,

I'm pleased to inform you that your manuscript has been deemed suitable for publication in PLOS ONE. Congratulations! Your manuscript is now being handed over to our production team.

Kind regards,

on behalf of

Dr. Benjamin Benzon

Academic Editor

PLOS ONE